# Dissection of Crop Metabolome Responses to Nitrogen, Phosphorus, Potassium, and Other Nutrient Deficiencies

**DOI:** 10.3390/ijms23169079

**Published:** 2022-08-13

**Authors:** Yingbin Xue, Shengnan Zhu, Rainer Schultze-Kraft, Guodao Liu, Zhijian Chen

**Affiliations:** 1College of Coastal Agricultural Science, Guangdong Ocean University, Zhanjiang 524088, China; 2Life Science and Technology School, Lingnan Normal University, Zhanjiang 524048, China; 3Alliance of Bioversity International and International Center for Tropical Agriculture, Cali 763537, Colombia; 4Institute of Tropical Crop Genetic Resources, Chinese Academy of Tropical Agricultural Sciences, Haikou 571101, China

**Keywords:** metabolome, crop metabolism, nutrient deficiency, metabolites

## Abstract

Crop growth and yield often face sophisticated environmental stresses, especially the low availability of mineral nutrients in soils, such as deficiencies of nitrogen, phosphorus, potassium, and others. Thus, it is of great importance to understand the mechanisms of crop response to mineral nutrient deficiencies, as a basis to contribute to genetic improvement and breeding of crop varieties with high nutrient efficiency for sustainable agriculture. With the advent of large-scale omics approaches, the metabolome based on mass spectrometry has been employed as a powerful and useful technique to dissect the biochemical, molecular, and genetic bases of metabolisms in many crops. Numerous metabolites have been demonstrated to play essential roles in plant growth and cellular stress response to nutrient limitations. Therefore, the purpose of this review was to summarize the recent advances in the dissection of crop metabolism responses to deficiencies of mineral nutrients, as well as the underlying adaptive mechanisms. This review is intended to provide insights into and perspectives on developing crop varieties with high nutrient efficiency through metabolite-based crop improvement.

## 1. Introduction

Along with the growing population, the demands for crop production have increased gradually, as reflected by the 85% increase predicted from 2013 to 2050 [1,2]. In different growth and development periods, crops often suffer many sophisticated environmental stresses. Among them, deficiencies of mineral nutrients, such as nitrogen (N), phosphorus (P), potassium (K), and others, are considered as the major constraints for crop growth and production [3,4]. For example, a 30–40% decrease in crop yield may occur under low P availability, while crop yields can drop by 10–40% under varying levels of N deficiency [4,5,6]. To maintain crop growth and yield under poor soil nutrient conditions in traditional agriculture, large amounts of chemical fertilizers are supplied to soils, but most of them are inevitably wasted due to low nutrient efficiency of crops and poor mobilization of nutrients in soils [7,8]. For example, N use efficiency of most plants is only 30–50%, resulting in 50–70% of the N fertilizer lost by denitrification, leaching, and volatilization [9]. Excess fertilizer amounts supplied to the soils lead to a waste of resources and increasing environmental issues, such as soil hardening, surface and groundwater contamination, and greenhouse gas emissions [10,11]. Therefore, developing crop varieties with high nutrient efficiency through genetic improvement is a critical approach to reconcile increased crop production with environmental sustainability.

To understand the adaptive mechanisms of crops to nutrient deficiency, a large number of nutrient-responsive genes or proteins have been identified and characterized through high-throughput omics techniques, such as genomics, transcriptomics, and proteomics [12,13]. Due to the changes in gene transcripts, protein levels, and enzyme activities not always being correlated, metabolites, the products of plant metabolism, are regarded as the readouts of plant growth or developmental status [14]. Thus, metabolomics, which is defined as qualitative and quantitative analysis of cellular metabolites based on mass spectrometry (MS) coupled to gas or liquid chromatography (GC or LC) and nuclear magnetic resonance (NMR) spectroscopy, has become an important complementary tool for functional genomics and system biology studies in plants [15]. 

More than 200,000 metabolites are estimated to be present in plants, which may have diverse functions in plant growth or cellular stress responses [14,16]. With the development of the accurate and large-scale detection of metabolites, metabolomics, including untargeted and targeted approaches, is now widely employed to identify differentially accumulated metabolites (DAMs) in response of crops to nutrient deficiencies. In this review, therefore, we mainly focus on recent advances in metabolomic dissection of crops in response to deficiencies of various mineral nutrients, including N, P, K, and other nutrients. This review also highlights the roles of key metabolites and the regulation of critical metabolic pathways during nutrient deficiency, with the intention to provide some insights into and perspectives on metabolite-based crop improvement.

## 2. Metabolisms Responsive to Nutrient Deficiencies in Crops

### 2.1. N Deficiency

As N is one of the most important macronutrients for crop growth and development, its deficiency severely decreases crop biomass, inhibits chlorophyll content, and disrupts photosynthesis and photorespiration, ultimately limiting crop yield [17,18,19]. A series of physiological and molecular mechanisms underlying crop adaptation to N deficiency have been demonstrated, such as coordinating carbon (C) and N metabolisms, regulating root architecture, modulating phytohormone signaling, enhancing N uptake and translocation, and accumulating stress tolerance-related compounds [8,20,21,22,23,24]. Since total N content and crop growth are affected by N limitation, metabolome analysis has been performed to identify N deficiency responsive metabolites and metabolic pathways, dissecting the adaptive mechanisms through regulation of metabolic profiles in many crops, such as rice (*Oryza sativa*), maize (*Zea mays*), wheat (*Triticum aestivum*), barley (*Hordeum vulgare*), soybean (*Glycine max*), tomato (*Solanum lycopersicum*), and rapeseed (*Brassica napus*) [25,26,27,28,29,30,31,32]. A summary of metabolome analyses of crops responses to N deficiency is presented in Table 1. Many of the identified DAMs can be integrated into specific metabolic pathways regulated by low-N stress (Figure 1).

N deficiency has been shown to significantly decrease photosynthetic rate [29,33]. Several photosynthesis-related genes and proteins have been found to be downregulated by N deficiency [12,25], which is closely related to the accumulation of carbohydrates under N deprivation [34]. A variety of sugars, including fructose, galactose, glucose, sucrose, and maltose, are markedly increased in N-deficient leaves of barley according to metabolome analysis [35]. A similar result has been reported in apple leaves, where several carbohydrates related to C metabolism, such as glucose-6-P, fructose-6-P, and glycerate-3-P, are increased by N deficiency [33]. The accumulation of carbohydrates is believed to act as a key signal to fine-tune the decrease in photosynthesis in plant leaves during N limitation. Consistent with the reduction in photosynthesis, the tricarboxylic acid (TCA) cycle is also inhibited in leaves during low-N stress [25,33,36]. For example, intermediate metabolites involved in the TCA cycle, such as 2-oxoglutarate, citrate, isocitrate, succinate, fumarate, and malate, are decreased in tomato leaves under N-deficient conditions [37].
ijms-23-09079-t001_Table 1Table 1Metabolome analyses of crop responses to N deficiency.Crop SpeciesTissueDuration of Treatment (days)MethodNumber of DAMsMain Changes in Metabolites or Metabolic PathwaysReferenceMaize (*Zea mays*) ^a^Leaves20/30GC–MS70 (in total)Decreasing most amino acids; increasing starch and secondary metabolites.[25]Tomato (*Solanum lycopersicum*)Leaves5/15LC/GC–MS28/34Decreasing amino acids and organic acids; increasing Fru-6-P, Glc-6-P, and sedoheptulose-7-P.[26]
Roots5/15LC/GC–MS28/34Decreasing amino acids and organic acids; increasing Fru-6-P, glucose, Glc-6-P, glycerate, pyruvate, ribulose, fructose, and sucrose.[26]Rice (*Oryza sativa*)Shoots5/15CE–TOF MS49/65Decreasing l-aspartate, l-phenylalanine, GABA, guanosine, adenine, and cytidine.[28]
Roots5/15CE–TOF MS59/73Decreasing nicotinamide, sorbitol-6P, glycero-3P, l-phenylalanine, GABA, citrulline, acetylserine, and histidinol.[28]
Root exudates5/15CE–TOF MS17/24Increasing glutarate, adipate, 2-hydroxyisobutyrate, succinate, 2-isopropylmalate, raffinose, and abscisate.[28]
Leaves30LC–ESI-MS/MS432Promoting TCA cycle to produce more energy and α-ketoglutarate.[29]Barley (*Hordeum vulgare*) ^a^Leaves1/3/6/9/12/15/18GC–MS51 (in total)Decreasing all major amino acids.[27]
Roots1/3/6/9/12/15/18GC–MS51 (in total)Increasing both minor and major amino acids at late stage.[27]
Shoots20GC–MS51Decreasing amino acids (glycine, asparagine, aspartic acid, glutamine, lysine, and threonine); increasing sugars (maltose, glucose, fructose, galactose, and psicose).[35]Barley (*Hordeum vulgare*) ^a^Roots20GC–MS49Decreasing amino acids (lysine, tyrosine, threonine, ornithine, and glutamine)[35]Soybean (*Glycine max*) ^a^Roots14GC–MS36/40Increasing soluble sugars and organic acids.[36]Wheat (*Triticum aestivum*)Grains25 days post anthesisGC–MS77Increasing ornithine, cysteine, aspartate, and tyrosine; promoting sugar accumulation.[31]Rapeseed (*Brassica napus*)Leaves14LC–ESI-MS/MS175Decreasing aspartic acid; increasing l-alanine.[32]Rapeseed (*Brassica napus*)Roots14LC–ESI-MS/MS166Increasing aspartic acid.[32]Apple (*Malus pumila*)Leaves30LC–ESI-MS/MS527Increasing ornithine, arginine, and asparagine.[33]
Roots30LC–ESI-MS/MS477Decreasing cinnamic acid, cyanidin-3-*O*-glucoside, and pelargonidin-3-*O*-glucoside[33]DAMs, differentially accumulated metabolites. ^a^ Two genotypes used in the studies.
Figure 1Metabolic pathways changes in crop leaves under N deficiency. N deficiency inhibits the N assimilation pathway and TCA cycle, resulting in large decreases in amino acids, while it accumulates tolerance-related metabolites for reactive oxygen species (ROS) scavenging. The accumulated and reduced metabolites are marked in red and blue, respectively. Abbreviations: P (phosphate), PEP (phosphoenolpyruvic acid), GABA (γ-aminobutyric acid), TCA (tricarboxylic acid), GS/GOGAT (glutamine synthetase/glutamate synthetase).
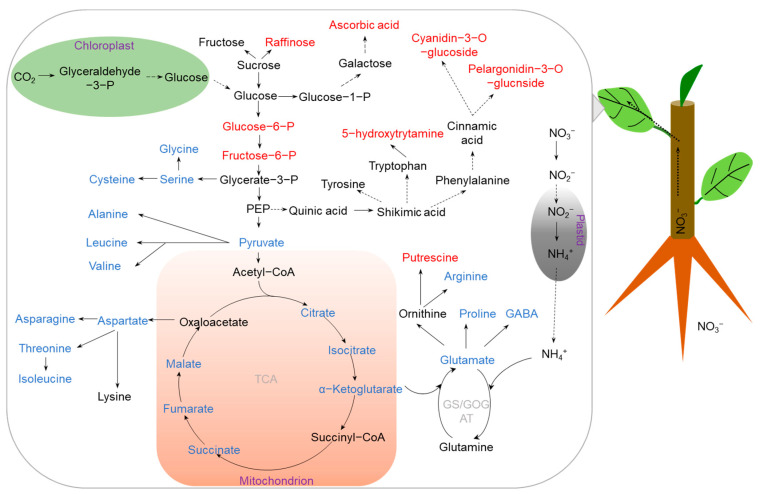



It has been demonstrated that N deficiency is bound to affect N metabolism. For example, the concentrations of free amino acids were decreased by 12.5% in leaves of rice exposed to low-N treatment [29]. In maize, a set of amino acids, such as glutamate, asparagine, alanine, serine, and glycine, were all decreased in leaves under N deficiency [25]. Similarly, under N-deficient conditions, most amino acids, including aspartic acid, lysine, glycine, threonine, asparagine, and glutamine, were decreased in barley leaves [35]. Interestingly, the decreased amino acid metabolites may be attributed to the downregulated glutamine synthetase (*GS*) and glutamine aminotransferase (*GOGAT*) genes, which are involved in the conversion of ammonium to amino acids [33]. Furthermore, integrated analyses of transcriptomics and metabolomics in rice showed that the NADH-dependent glutamate synthase (*OsGLT1*) gene is downregulated by N deficiency, which is consistent with the decreases in glutamate concentration [29]. Similar results have also been obtained in leaves of soybean and tomato where amino acids were decreased by N deficiency [26,30]. The reduction in amino-acid levels under low-N stress is considered as an energy-saving strategy. On the basis of the above results, it is reasonable to propose that a crop can adjust the balance between C and N metabolism to avoid metabolic inefficiencies and maintain crop growth under N deprivation. 

Similar to other abiotic stresses, N deficiency also causes the generation of reactive oxygen species (ROS), resulting in lipid peroxidation and triggering oxidative stress in crops if not well scavenged. For example, the content of H_2_O_2_ is significantly increased in apple leaves subjected to low-N treatment, generating oxidative stress [33]. In addition to induce the activities of antioxidant enzymes to eliminate ROS damage, the other adaptive change that alleviates oxidative stress during N deficiency in plants is the accumulation of stress tolerance-related metabolites. Among these metabolites, galactinol, raffinose, sugar alcohols, ascorbic acid, and polyamines have been considered as ROS scavengers [38,39]. It was reported that ascorbic acid, putrescine, and 5-hydroxytryptamine were greatly accumulated in barley shoots [35], which are beneficial for the tolerance of barley to low-N stress. Secondary metabolites are proposed to be important resistance substances produced by plants during long-term adaptation to environmental stress. Metabolome studies showed that flavonoid-related metabolites, such as cinnamic acid, dihydroquercetin, pelargonidin-3-*O*-glucoside, and cyanidin-3-*O*-glucoside, were increased by N deficiency in apple [33], which is likely to protect cells from oxidative stress damage. Furthermore, under N limitation, β-alanine levels were found to be increased in leaves of rapeseed [32]. Increases in the levels of osmoprotectants, such as β-alanine, proline, and γ-aminobutyric acid (GABA), are generally associated with enhanced low-N stress tolerance in plants [40], but the exact roles of these metabolites in different crops remain to be investigated.

Another strategy for increasing low-N stress tolerance can be achieved by promoting root elongation under N deficiency. Thus, metabolic profile changes in roots can reveal the mechanisms underlying adaptation of a crop to N deficiency. For example, plant hormones are found to play an important role in regulating root growth under low-N stress [32]. The concentrations of gibberellic acid (GA) in rapeseed roots were significantly increased under N deficiency, which may contribute to promoting root growth [32]. In addition to phytohormones, increasing C partitions to roots is also necessary to increase root growth [41,42]. In contrast to leaves, the levels of metabolites involved in the TCA cycle were increased in apple and soybean roots under N deficiency [30,33], which may promote root growth through enhancing energy accumulation under N-deficient conditions. On the contrary, the contents of alanine, aspartic acid, isoleucine, serine, and threonine were found to be decreased in low-N-tolerant soybean roots, indicating that low-N-tolerant soybean may adapt to N deficiency by reducing energy consumption [30]. Malate, related to the TCA cycle, was found to be increased in roots under N deficiency [33]. Since dehydrogenation of malate is accompanied by the generation of NADH, which is an important antioxidant, the increased malate concentration in roots is considered as an adaptive mechanism of plant tolerance to N deficiency by an enhanced antioxidant status [33]. An additional study in soybean showed that the accumulation of malate in roots could also stimulate nitrate uptake under N deficiency [43]. Furthermore, secondary metabolites, such as salicylic acid (SA) and catechol, were increased in soybean roots under N deprivation [30]. SA was found to be involved in increasing N use efficiency of isolated cucumber (*Cucumis sativus*) cotyledons [44]. Moreover, the shikimate metabolic pathway-related compounds phenylalanine, shikimic acid, SA, naringin, and neohesperidin also increased in soybean roots during N deficiency [30], which may contribute to the synthesis of aromatic amino acids, plant hormones, and a variety of important active secondary metabolites, increasing tolerance to stress conditions [45,46]. Furthermore, the levels of raffinose and galactitol in roots were higher than those in shoots of barley [35]; the authors concluded that roots were more affected by low-N stress than shoots. A comparison of amino-acid metabolites in common soybean with the low-N-tolerant soybean genotype Tongyu06311 showed that proline was accumulated in roots of the low-N-tolerant soybean genotype Tongyu06311, which is probably beneficial for soybean adapted to low-N stress [30]. Thus, metabolism adjustments are essential for crops in response to N deficiency.

### 2.2. P Deficiency

P is a key component of nucleic acids, proteins, and membrane lipids, and it is essential for many biological processes in plants [13,47,48]. Low P availability in soils is a major constraint for crop production. In past decades, there have been large advances in dissecting the mechanisms of plant adaptation to P deficiency including physiological and biochemical responses. Plants have developed a variety of adaptive strategies, such as changing root architecture and morphology, increasing the secretion of organic acids, and developing a bypass pathway for recycling internal P [12,49,50]. Metabolome analysis has also been widely conducted to investigate the metabolite-based low-P tolerance mechanisms in crops, such as soybean, quinoa (*Chenopodium quinoa*), common bean (*Phaseolus vulgaris*), tomato, and oats (*Avena sativa*) [26,51,52,53,54,55]. To date, numerous metabolites have been identified to be involved in the responses of crops to P deficiency. A summary of metabolome analysis and identified DAMs is presented in Table 2. The DAMs can be integrated into specific pathways associated with lipids, flavonoids, amino acids, and nucleotide metabolisms, shedding light on the changes in crop responses to low-P stress (Figure 2). These findings provide major insights into understanding the mechanisms of low-P stress tolerance through metabolic modulation.

Modifying root growth and increasing the root-to-shoot ratio are key adaptive mechanisms to enhance phosphate (Pi) acquisition efficiency for plants under low-P stress. Transcriptomic and proteomic analyses have been conducted to identify key genes or proteins involved in the regulation of root architecture and morphology in response to P deficiency [54]. Metabolites involved in root development regulation have also been identified through a metabolomic approach [26,51,53,54]. Both C and N metabolisms have been reported to be modulated in response of crops to P deficiency. Most amino-acid metabolites, including asparagine, lysine, histidine, ornithine, isoleucine, leucine, and arginine, were found to be accumulated in P-deprived roots of several crops, such as common bean, tomato, and soybean [26,51,54]. Furthermore, it was found that the increase in amino-acid concentration may be due to the upregulation of protein degradation-related genes and the downregulation of protein synthesis-related genes under P deficiency [56,57]. During low-P stress, plants can increase C distribution to the root system, thereby increasing the root-to-shoot ratio and regulating the root system morphology. Significant increases in maltose, sucrose, raffinose, and 6-kestose were observed in barley roots under 17 days of low-P treatment [58]; the authors considered this an adaptive mechanism of plants by promoting root growth through regulating C allocation. In addition, sugar has been documented to be an important sensor for the Pi starvation response; the expression of phosphate starvation-induced (PSI) genes was found to be regulated by sugar limitation [59]. Thus, increases in sugar levels in roots may induce the expression of PSI genes, regulating plant growth under low-P stress. However, further characterization of sugar and PSI genes is needed to confirm their exact roles in low-P stress tolerance via regulating C allocation in plants.

In addition to root growth regulation for acquiring Pi, crop roots can exudate organic acids into the rhizosphere to promote solubilization of fixed Pi [32,60]. It has been found that organic acids have an important role in the response of plants to Pi starvation. For example, metabolome analysis of the exudates from rice roots revealed that organic acids, such as 2,6-diaminopimelate, 3-dehydroshikimate, fumarate, hypoxanthine, and d-galacturonate, were increased by P deficiency [28], which may contribute to the mobilization of insoluble soil P, as suggested by the authors. Furthermore, significant increases in the exudation of malic, oxalic, and succinic acids were observed in the P-efficient wheat genotype RAC875 [61]. On the other hand, metabolome analysis has shown that internal organic acids in roots are also affected by P deficiency. The levels of organic acids, such as tartaric acid and 2,4-dihydroxybutanoic acid, in roots of common bean were found to be decreased during low-P stress [52]. Similar results were also obtained in barley roots exposed to low-P treatment, where the levels of several organic acids, including α-ketoglutarate, succinate, fumarate, and malate, were reduced [58]. Therefore, organic acids secreted to the rhizosphere may lead to the reduction in organic acids in roots under P deficiency. An increase in organic acid exudation from roots is one of the important physiological mechanisms for crops increasing Pi utilization from soils. 

On the other hand, promoting the remobilization of internal P resources, such as phosphorylated metabolites, nucleic acids, and phospholipids, which are well known as the largest P pool in plants [62], is necessary for crop adaptation to P deficiency. Under P-limited conditions, the levels of phosphorylated metabolites were reported to be decreased in soybean roots, including *sn*-glycero-3-phosphocholine, *O*-phosphocholine, deoxyribose 5-phosphate, *O*-phosphorylethanolamine, and dl-glyceraldehyde 3-phosphate [54]. Similar results were also found in oats where glucose-6-phosphate and myo-inositol phosphate were dramatically decreased in P-deficient roots [53]. Moreover, nucleotides, such as adenosine 3′-monophosphate, inosine 5′-monophosphate, guanosine 5′-monophosphate, uridine 5′-diphospho-D-glucose, guanosine monophosphate, adenosine 5′-monophosphate, deoxyribose 5-phosphate, cytidine 5′-monophosphate, uridine 5′-monophosphate, and guanosine 3′,5′-cyclic monophosphate, were decreased by Pi starvation in soybean roots [54]. Decreases in nucleic acid concentration were also observed in white lupin under Pi starvation [63]. The regulation of the synthesis and/or degradation of nucleotides is likely to help a crop cope with P deficiency. Recently, a key gene, *DNA polymerase delta 1* (*DPD1*), involved in organelle DNA degradation for improving P use efficiency, was characterized in *Arabidopsis* [64]. Several *DPD1* homologs in soybean were also found to be upregulated in roots under P deficiency [54]. These results support the hypothesis that changes in nucleotide metabolism are beneficial for increasing internal P remobilization, thereby improving P utilization efficiency. Furthermore, lipid-related metabolites such as glycerophospholipids were found to be decreased in responses of crops to P deficiency [54,65]. For example, in soybean roots, *sn*-glycero-3-phosphocholine, *O*-phosphocholine, and several glycerophospholipids, all of which are involved in remodeling membrane lipids, were decreased under P-deficient conditions [54]. Replacing phospholipids with sulfolipids or galactolipids in bio-membranes can also help plant tolerance to low-P stress; this deserves further investigation.
ijms-23-09079-t002_Table 2Table 2Metabolome analyses of crop responses to P deficiency.Crop SpeciesTissueDuration of Treatment (days)MethodNumber of DAMsMain Changes in Metabolites or Metabolic PathwaysReferenceTomato (*Solanum lycopersicum*)Leaves5/15LC/GC–MS17/30Decreasing soluble sugars.[26]
Roots5/15LC/GC–MS29/30Decreasing soluble sugars; increasing amino acids and organic acids.[26]Rice (*Oryza sativa*)Shoots5/15CE–TOF MS26/38Decreasing l-aspartate, l-phenylalanine, GABA, guanosine, adenine, and cytidine.[28]
Roots5/15CE–TOF MS33/8Decreasing *trans*-zeatin, citrate, and d-glucosamine.[28]
Root exudates5/15CE–TOF MS18/12Increasing cytosine, hypoxanthine, nicotinate, choline, 1,4-butanediamine, creatine, 2,6-diaminopimelate, 3-dehydroshikimate, galactosamine, fumarate, glycerate, and glutamate.[28]Common bean (*Phaseolus vulgaris*)Roots21GC–MS42Increasing polyols and sugars.[51]
Nodules21GC–MS45Increasing organic and polyhydroxy acids.[52]Oats (*Avena sativa*)Roots10GC–MS30Decreasing phosphorylated metabolites; increasing citric acid and malic acid.[53]Soybean (*Glycine max*)Roots12LC–ESI-MS/MS155Decreasing phosphorylated lipids and nucleic acids.[54]Quinoa (*Chenopodium quinoa*) ^a^Shoots30UPLC–MS/MS149Decreasing dihydroxyacetone phosphate, 3-phospho-d-glyceric acid, glucose-1-phosphate, and uridine diphospho-d-glucose[55]Barley (*Hordeum vulgare*)Shoots20GC–MS51Decreasing phosphorus-containing compounds (glucose-6-phosphate, mannose-6-phosphate, and glycerol-3-phosphate).[35]
Roots20GC–MS49Increasing sugars (fructose, glucose, and sucrose) and organic acids (citric acid and malic acid).[35]
Shoots10/17GC–MS22/38Decreasing glucose-6-P, fructose-6-P, glycerol-3-P, and inositol-1-P.[58]
Roots10/17GC–MS7/42Decreasing succinic acid and fumaric acid.[58]Wheat (*Triticum aestivum*)Leaves/roots28GC–MSndDecreasing glycerol-3-P in roots; increasing raffinose and 1-kestose in roots and aspartate, glutamine, and alanine in leaves.[61]White lupin (*Lupinus albus*)Shoots14/22GC–MSndDecreasing fructose, glucose, and sucrose after 14 days of treatment.[63]
Non-cluster roots14/22GC–MSndDecreasing phosphorylated metabolites; increasing organic acids and several shikimate pathway products.[63]DAMs, differentially accumulated metabolites; nd, not described in the studies. ^a^ Two genotypes used in the studies.


In contrast to roots, increased accumulation of sucrose, maltose, raffinose, and 6-kestose was observed mainly in shoots of barley growing under moderately P-deficient conditions [58], indicating that barley roots are less sensitive to Pi starvation. Furthermore, amino acids in legume nodules are also significantly affected by P deficiency. For example, five out of 10 amino-acid metabolites were decreased, whereas three out of 10 amino-acid metabolites were increased in nodules of common bean [52]. N metabolism-related metabolites, including spermidine, putrescine, urea, glycine, serine, glutamine, and threonine, were reduced in nodules of common bean under P deficiency, which may lead to a decrease in symbiotic nitrogen fixation [52]. However, the mechanism of metabolite changes in nodules under low-P stress requires to be studied further. 

### 2.3. K Deficiency

Among the macronutrients, K plays essential roles in plant growth and development as a major cation or as a cofactor of various enzymes. Unlike N and P, K is not a part of organic compounds, but plays important roles in many physiological and biochemical processes, such as enzyme activation, ion homeostasis, osmoregulation, and protein synthesis [66,67]. Generally, the availability of K in soils is limited, which has become a limiting factor for sustainable production of cultivated crops [68]. Recently, metabolomic approaches have been applied to dissect the mechanism of crop tolerance to K deficiency (Table 3); examples include tomato (*Solanum lycopersicum*), sunflower (*Helianthus annuus*), barley (*Hordeum vulgare*), rapeseed (*Brassica napus*), and peanut (*Arachis hypogaea*) [26,35,69,70,71,72]. Many of the identified DAMs can be integrated into specific metabolic pathways regulated by K deficiency stress (Figure 3). 

It is generally believed that carbohydrate metabolism not only is an important energy source for plants, but also plays a vital role in protein and lipid metabolisms [73]. Increases in the content of sugars, such as glucose, sucrose, and fructose, are suggested to be associated with plants in response to various stresses, including K deficiency [74]. Sugar levels have been reported to be increased in both leaves and roots of barley under K deficiency [35,70]. Accumulation of sucrose was also found in tomato roots under low-K stress [26]. Furthermore, low-K-tolerant barley genotypes seemed to accumulate more sugars in both leaves and roots than low-K-sensitive barley genotypes [70], indicating that increasing sugar accumulation is critical for barley adaptation to low-K stress. In addition, sucrose is an important signaling molecule that is transferred from leaves to roots, regulating root growth in response to nutrient stress [25,52]. Since K is involved in the loading of sucrose to the phloem, availability of K seriously affects the transport of sucrose from leaves to roots [75,76]. Therefore, under K-deficient conditions, sucrose in roots is not only an important substance for low-K tolerance, but also a key indicator to screen crops for tolerance to K limitation. It has been documented that N metabolism is affected by K deficiency; according to metabolome analysis, amino acids in leaves and roots of barley were increased during K limitation [70]. Metabolomic analysis also showed that tryptophan, guanidineacetic acid, asparagine, alanine, ornithine, and histidine were all increased in K-deficient wheat roots, while citric acid, glutamic acid, and GABA were decreased [77]. Interestingly, most of the increased amino acids were positively charged, whereas the negatively charged amino acids were reduced in both leaves and roots of barley [70]. Since K deficiency could lead to electric charge imbalance, it is important to maintain charge balance in plant cells to cope with low-K stress. The phenylpropanoid metabolic pathway is one of the most important secondary metabolic pathways in plants [78]. Within this pathway, l-phenylalanine can be catalyzed into *trans*-Cinnamic acid, which is a key substrate for the synthesis of flavonoids, lignin, and alkaloids [79]. Metabolome analysis revealed that, under K-deficient conditions, l-phenylalanine levels in a low-K-tolerant barley genotype were higher than those in a low-K-sensitive barley genotype [70], suggesting that regulation of the phenylpropanoid metabolic pathway can contribute to barley coping with low-K stress.
Figure 3Metabolic pathways regulated by K deficiency in crops. K deficiency affects diverse pathways, including N metabolism, TCA cycle, glycolysis, shikimic acid pathway, and secondary metabolic pathways, accumulating stress-tolerant metabolites and phytohormones. The accumulated and reduced metabolites are marked in red and blue, respectively. Abbreviations: SA (salicylic acid), JA (jasmonic acid), ABA (abscisic acid), GABA (γ-aminobutyric acid), TCA (tricarboxylic acid), GS/GOGAT (glutamine synthetase/glutamate synthetase).
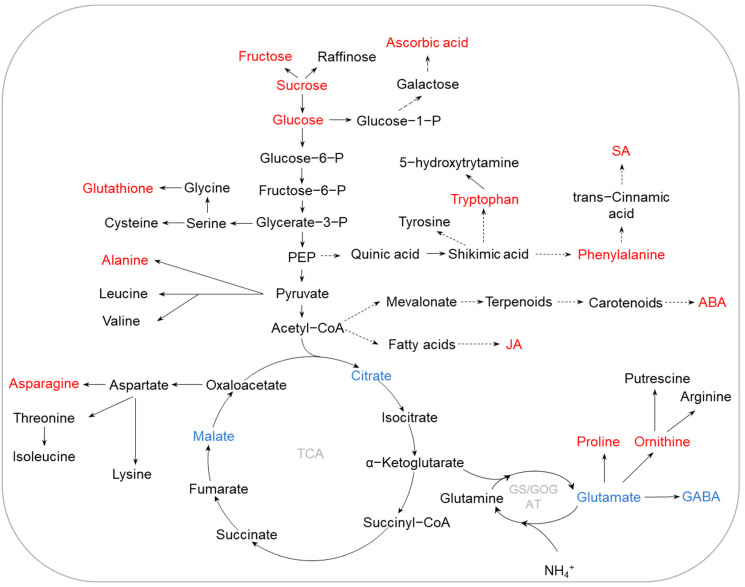



K deficiency also causes an excess accumulation of reactive oxygen species (ROS), resulting in oxidative stress in plants [80]. Thus, increasing the concentration of antioxidant metabolites is a vital stress tolerance strategy for plants dealing with K deprivation. The accumulation of compatible solutes, such as proline, soluble sugars, amino acids, and polyols, plays an important role in osmotic adjustment [81]. Among them, proline is regarded as an important antioxidant for stress tolerance [82]. There is evidence that K deficiency increases the concentration of proline in both leaves and roots of barley; for example, leaves of the low-K-tolerant cultivar XZ153 contained higher proline levels than those of the sensitive cultivar XZ141 [70]. Similarly, increases in proline concentration were observed in K-deprived leaves and roots of peanut [72]. In addition, ascorbic acid is an important antioxidant protecting cell membrane permeability [83]. The concentration of ascorbic acid in barley roots was found to be increased during low-K stress, especially in the low-K-tolerant cultivar XZ153. Furthermore, ascorbic acid concentrations were increased in leaves of the low-K-tolerant barley cultivar XZ153, but decreased in the low-K-sensitive barley cultivar XZ141 [70]. In addition, glutathione is also a key antioxidant involved in scavenging ROS via the GSH-ascorbate cycle [84]. Metabolome analysis showed that the content of glutathione was increased in roots of the low-K-tolerant wheat cultivar KN9204 but not in low-K-sensitive cultivar BN207 [77]. Thus, it is reasonable to propose that antioxidant metabolites, such as proline, ascorbic acid, and glutathione, are important metabolites for crop adaptation to K deficiency, although further investigation is required.

Phytohormones are small endogenous signaling molecules that participate in regulating plant growth and development in various life stages and stress conditions. Metabolites related to phytohormones, such as abscisic acid (ABA), jasmonic acid (JA), and SA, are regulated by K deficiency. ABA is well known as a stress signal in response to drought, salinity, and nutrient limitation [85]. It can maintain the water relation by regulating stomatal conductance and plant metabolism [86]. JA is involved in abiotic stress through activation of antioxidant systems, synthesis of amino acids and sugars, and regulation of stomatal opening and closing [87]. SA is involved in protecting membrane integrity and modulating abundance of protein associated with secondary metabolites [72]. It was shown that, in both leaves and roots of peanut, K deficiency increased the levels of ABA [72]. Similarly, JA concentration in leaves of peanut also increased during low-K stress [72]. Unlike ABA and JA, SA concentration increased in leaves of peanut but decreased in roots under K-limited conditions [72]. Therefore, considering the importance of phytohormones in plant growth, it is reasonable to suggest that ABA, JA, SA, and other phytohormones are important molecules for low-K stress tolerance.

### 2.4. Other Nutrient Deficiencies

Despite the advances in identifying various metabolites and metabolic pathways responding to N, P, and K deficiency, little attention has been given to metabolic changes in response of crops to deficiencies of other essential nutrients, such as magnesium (Mg), iron (Fe), zinc (Zn), sulfur (S), and boron (B) (Table 3). 

Mg is an important component of chlorophyll and a cofactor for enzymes participating in many physiological processes [88]. It has been reported that Mg deficiency leads to large differentiated metabolic processes in source and sink tissues. For example, Mg deficiency led to leaf-specific accumulation of amino-acid metabolites in soybean, such as phenylalanine, asparagine, leucine, isoleucine, glycine, glutamine, and serine; in contrast, root-specific depletion of pyruvic acid, citrate, 2-keto-glutaric acid, succinic acid, fumaric acid, and malate were observed under Mg deficiency [89]. Mg deficiency also impaired C allocation in soybean, as reflected by significant increases in carbohydrates, such as starch, sucrose, glucose, and fructose in leaves, and moderate decreases in sucrose and starch in roots [89]. These results suggest that reprogramming of distinct C and N metabolisms may occur in the response of soybean leaves and roots to Mg limitation. 

Fe is the fourth most common element in the Earth’s crust, and it is easily fixed into insoluble Fe^3+^ precipitates, leading to low availability for plants [90]. Fe limitation affects several metabolic processes, such as photosynthesis and respiration, as well as leads to an increase in ROS [91]. In rice, glycolysis and respiration-related metabolites, such as 3-P-glycerate, 3-P-glycerate derivatives, branched-chain amino acids, and pyruvate derivatives, were found to be increased in roots during low-Fe stress [91]. Furthermore, an increase in phytosiderophore 2′-deoxymugineic acid was observed in rice roots under Fe deficiency [91]. These results suggest that changes in C and energy metabolisms and increasing 2′-deoxymugineic acid secretion are important adaptive mechanisms of rice dealing with Fe deficiency. In addition, in leaves of the betel palm (*Areca catechu*), significant increases in naringenin, butin, and hesperetin but decreases in xanthohumol, purine, and N-p-coumaroylspermidine were observed under Fe deficiency [92], suggesting that regulating biosynthesis of flavonoids and flavonols is an important adaptive strategy for the betel palm in response to Fe deficiency.
ijms-23-09079-t003_Table 3Table 3Metabolome analyses of crop responses to other nutrient deficiencies.NutrientCrop SpeciesTissueDuration of Treatment (d)MethodNumber of DAMsMain Changes of Metabolites or Metabolic PathwaysReferencePotassiumTomato (*Solanum lycopersicum*)Leaves5/15LC/GC–MS30/28Decreasing organic acids and amino acids.[26]

Roots5/15LC/GC–MS32/29Accumulating soluble sugars and amino acids.[26]
Barley (*Hordeum vulgare*) ^a^Shoots20GC–MS51Increasing monosaccharides (fructose, galactose, and glucose), disaccharides (sucrose and maltose), and polysaccharide (psicose).[35]

Roots20GC–MS49Increasing putrescine and 5-hydroxytryptamine.[35]

Leaves/roots16GC–MS57 (in total)Decreasing negatively charged amino acids (Asp and Glu) and most organic acids; increasing positively charged amino acids (Lys and Gln).[70]
Sunflower (*Helianthus annuus*)Leaves/roots14GC–MSndDecreasing glycerol 3-phosphate and fructose 6-phosphate; increasing citrate, aconitate, malate, fumarate, and putrescine.[69]
Rapeseed (*Brassica napus*) ^b^Leaves45LC–MSndIncreasing citric acid, arginine, and asparagine.[71]
Peanut (*Arachis hypogaea*)Leaves/roots15GC–MSndDecreasing aspartic acid and glutamic acid; increasing lysine, histidine, and arginine[72]
Wheat (*Triticum aestivum*) ^a^Roots14UPLC–ESI-MS/MS162Decreasing more amino acids in K-sensitive genotype BN207; increasing more amino acids in K-tolerant genotype KN9204. [77]MagnesiumSoybean (*Glycine max*)Leaves4/8GC–MS5/26Decreasing methylmalonic acid; increasing phenylalanine, carbon allocation, and respiration metabolism (e.g., sucrose, glucose, and fructose).[89]

Roots4/8GC–MS3/16Decreasing urea and TCA cycle; increasing glutamine and allantoic acid. [89]IronRice (*Oryza sativa*)Roots7LC–MSndIncreasing amino acids related to α-ketoglutarate family (proline, histidine, and glutamine).[91]
Betel palm (*Areca catechu*)Leaves28LC–MS106Increasing organic acids and flavonoids.[92]ZincTea (*Camellia sinensis*)Leaves120LC–MS10Decreasing fructose-6-phosphate, digalactosylglycerol, and 2-*O*-glycerol-beta-d-galactopyranoside; increasing caffeine and catechin gallate.[93]SulfurLettuce (*Lactuca sativa*) ^a^Leaves42LC–MS14Increasing caffeoyl derivatives, caffeic acid hexose, 5-caffeoylquinic acid (5-OCQA), quercetin, and luteolin glucoside derivatives.[94]BoronAlfalfa (*Medicago sativa*)Flowers7GC–MS19Increasing large sugars.[95]

SeedsUntil harvestGC–MS13Increasing sugars and phenolic compounds.[95]DAMs, differentially accumulated metabolites; nd, not described in the studies. ^a^ Two genotypes used in the studies; ^b^ three different degrees of K deficiency. nd, not described in the studies.


In tea (*Camellia sinensis*) plants, Zn deficiency reduced the contents of two secondary metabolites, four carbohydrate metabolites, and four nitrogenous metabolites in leaves [93], indicating that tea plants respond to Zn-deficient stress through regulating carbohydrate, nitrogenous, and secondary metabolisms. Recently, several secondary metabolites, such as sesquiterpene lactones, caffeoyl derivatives, caffeic acid hexose, 5-caffeoylquinic acid, quercetin, and luteolin glucoside derivatives, were found to be regulated by S deficiency in leaves of lettuce (*Lactuca sativa*) [94]. Furthermore, in alfalfa (*Medicago sativa*), B deficiency increased the accumulation of sugars and phenolic compounds in flowers and seeds, respectively, which may cause abscission or abortion of reproductive organs [95]. 

Although the results above provide some useful information on the changes in metabolic profiles of crops in response to deficiencies of Mg, Zn, Fe, S, and B, more studies in these areas are needed to increase our understanding of the metabolic mechanisms of crop adaptation.

## 3. Conclusions and Perspectives

Nutrient deficiency directly limits crop growth and production. With the rapid development of analytical detection technology and bioinformatics, metabolomics has become one of the important technologies in systems biology research to dissect metabolic profile responses of crops to nutrient stress. This review summarized the advances of crop metabolism responses to deficiencies of mineral nutrients and discussed these responses and the underlying adaptive mechanisms. N deficiency seems to impair the whole plant growth, as reflected by decreased N assimilation and TCA cycle, as well as a reduction in most amino acids, which is considered as an energy-saving strategy for tolerance to low-N stress. On the other hand, N deficiency often causes oxidative stress in plants; thus, several stress tolerance-related metabolites, such as galactinol, raffinose, sugar alcohol, and ascorbic acid, are accumulated under N-deficient conditions, contributing to ROS scavenging [38,39]. Regarding low P availability, N and C metabolisms are also affected by Pi deprivation, along with the TCA cycle and membrane phospholipid metabolism. This can be considered as fine-tuning to improve P efficiency in plants. For example, increases in sucrose and amino acids in roots seem to support the enlargement of roots. In addition, the reduction in organic acid metabolites may be attributed to the production of root exudates to mobilize soil P. Furthermore, the reduction in phospholipid metabolites, which are important sources of organic P, may contribute to P reutilization. Unlike N and P, K is not a component of most metabolites, and metabolism changes caused by K deficiency may be helpful for tolerance to osmotic oxidative stresses. N metabolism is also regulated by K deficiency, while the changes in amino acids (e.g., glutathione) may also relate to oxidative stress. Furthermore, several secondary metabolic pathways obviously change under K-deficient conditions, including the phenylpropanoid pathway, where accumulated phenylalanine can be converted into some secondary metabolites and salicylic acid, which are critical for stress tolerance. While accumulation of sugar metabolites is observed under N, P, and K deficiency, the increases in soluble sugars may contribute to maintain osmotic homeostasis during nutrient deficiencies. Therefore, both common and specific metabolites or metabolic pathways can play a part in crop responses to nutrient deficiency. Although more studies are needed, some key clues indicate that regulation of C, N, and energy metabolisms is important for the responses of crops to nutrient deficiencies, especially regarding macroelements. Elucidating the biosynthesis and regulation of crop metabolites during nutrient deficiency can largely increase our understanding of how plants acquire and utilize mineral nutrients under the fluctuating levels of nutrients in soils. 

Although significant advances in the diverse detection platforms, such as GC–MS, LC–MS, and capillary electrophoresis–mass spectrometry (CE–MS), have been used in metabolomic analysis, individual platforms are unable to cover all metabolites in plants [96], since the number of identified metabolites varies greatly across different techniques. Thus, making full use of the advantages of different detection platforms, multiplatform detection should be used for comprehensive metabolomic analysis. Although metabolomics data reveal various metabolic pathways regulated by nutrient stress, it remains hard to know whether a metabolic pathway is up- or downregulated, because most changed metabolites may be associated with two or more pathways. 

The changes in metabolism pathways are usually caused by a number of related functional transcripts rather than individual transcripts. Thus, it is important to integrate transcriptomics and metabolomics to identify the response of key genes or pathways to nutrient deficiency. On the other hand, changes of transcripts may not always correlate to enzyme activities; thus, proteomics can be used to identify key proteins or enzymes. It is also important for a better correlation of the changes between metabolites with genes and proteins, as well as crop growth and development. In consequence, future work is required to integrate analyses of transcriptomics, proteomics, and metabolomics to dissect the mechanisms underlying crop response to nutrient deficiency. 

Furthermore, as an important bridge between genome and phenome, metabolite-based genome-wide association study (mGWAS) has recently been used in interactive functional genomics and metabolomics to understand the genetic bases of plant metabolism [14,97]. The mGWAS approach is performed to identify key genes involved in specific metabolic pathways in crops. For example, in wheat, several candidate genes were identified as being involved in the flavonoid decoration pathway through mGWAS [98]. Using the mGWAS approach, a genetic network of chlorogenic acid biosynthesis in *Populus tomentosa* was constructed on the basis of six causal genes [99]. Similar results were also reported in barley for UV-B protection through the regulation of the phenylpropanoid pathway [100]. However, available information about mGWAS used for dissecting mechanisms underlying crop responses to nutrient deficiency is scarce. It is important to identify the critical genes participating in specific metabolic pathways through integration of mGWAS and other omics approaches, which could be used to develop high-nutrient-efficiency crop varieties through genetic improvement in future.

## Figures and Tables

**Figure 2 ijms-23-09079-f002:**
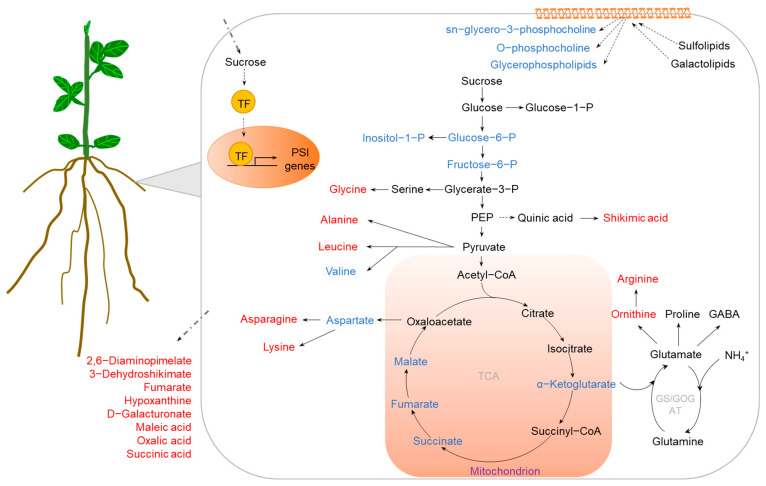
Metabolic pathways regulated by P deficiency in crop roots. The specific changes in metabolites involve organic acids and phospholipids, which may be related to external Pi acquisition and internal P reutilization, respectively. The accumulated and reduced metabolites are marked in red and blue, respectively. Abbreviations: PSI (phosphate starvation-induced), P (phosphate), PEP (phosphoenolpyruvic acid), GABA (γ-aminobutyric acid), TCA (tricarboxylic acid), GS/GOGAT (glutamine synthetase/glutamate synthetase).

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
