# Peer review of "Dissection of Crop Metabolome Responses to Nitrogen, Phosphorus, Potassium, and Other Nutrient Deficiencies"

_ijms, 2022, doi:10.3390/ijms23169079_

Round 1

Reviewer 1 Report

Type of manuscript: ReviewTitle: Dissection of Crop Metabolome Responses to Nutrient DeficienciesJournal: International Journal of Molecular Science

The article is well-written and in line with the scope of the journal. The authors review the metabolic response of plant crops to nutrient deficiencies, specifically “macroelements”, N, P and K, and barely to “micronutrients” Mg, Fe, Zn, S and B. Therefore, the tittle must be corrected to reflect that elemental perspective is reviewed specifically.

Figure 1 and 2 and too small and very difficult to follow for the reader since too much info is contained in both of them, the font size must be increased and maybe the content summarized as well. The content of the figures is broadly known and this reviewer does not find the link in between macroelement deficiencies and the impact on changes within the metabolic pathways described in the figures. Only N and P are pictured, it is understandable since N and P are part of important organic molecules for the plant metabolism, but this reviewer miss a figure also elucidating the role of K as key inorganic nutrient.

Tables 1, 2 and 3 are very little informative. They just collect a number of references and the analytical technique used for the determination of metabolites. This should be corrected by for instance including some info of the research conducted.

Besides the authors, in several occasions, reflect to the “omics” time as an opportunity to deep on the state-of-the-art, but I miss useful information about for instance genetic tools available for genome mining, metabolome mining and so on that can provide useful data to perform holistic studies.

Therefore, the authors must implement a major revision before this article is accepted for publication in ISMS.

Author Response

  1. The article is well-written and in line with the scope of the journal. The authors review the metabolic response of plant crops to nutrient deficiencies, specifically “macroelements”, N, P and K, and barely to “micronutrients” Mg, Fe, Zn, S and B. Therefore, the tittle must be corrected to reflect that elemental perspective is reviewed specifically.

Answer: We highly appreciate that you took time to clearly spell out the major issues of the manuscript we need to address. We have modified the title as “Dissection of Crop Metabolome Responses to Nitrogen, Phosphorus, Potassium and Other Nutrient Deficiencies.” Please check this in the revised manuscript.

  1. Figure 1 and 2 and too small and very difficult to follow for the reader since too much info is contained in both of them, the font size must be increased and maybe the content summarized as well. The content of the figures is broadly known and this reviewer does not find the link in between macroelement deficiencies and the impact on changes within the metabolic pathways described in the figures. Only N and P are pictured, it is understandable since N and P are part of important organic molecules for the plant metabolism, but this reviewer miss a figure also elucidating the role of K as key inorganic nutrient.

Answer: Thank you for your valuable comments and suggestions. We have added additional descriptions to discuss the link between macroelement deficiencies and the impact on changes within the metabolic pathways. Namely, N deficiency seems to impair the whole plant growth, as reflected by decreased N assimilation, TCA cycle and reduction of most amino acids (Figure 1), which is considered as an energy-saving strategy for low-N stress tolerance. On the other hand, N deficiency often causes oxidative stress in plants, and thus several stress tolerance-related metabolites, such as galactinol, raffinose, sugar alcohol and ascorbic acid, are accumulated under N-deficient condition (Figure 1), contributing to ROS scavenging [38,39]. Regarding low P availability, N and C metabolisms are also affected by Pi deprivation, along with the TCA cycle and membrane phospholipid metabolism. This can be considered as a fine tune-up to improve P efficiency in plants. For example, increases of sucrose and amino acids in roots seem to support the enlargement of roots. In addition, the reduction of organic acid metabolites may be attributed to the production of root exudates to mobilize soil P. Besides, the reduction of phospholipid metabolites, which are important sources of organic P, may contribute to P-reutilization (Figure 2). Different from N and P, K is not a component of most metabolites, and changes of metabolisms caused by K deficiency may be helpful for tolerance to osmotic and oxidative stresses (Figure 3). N metabolism is also regulated by K deficiency, but the changes of amino acids (e.g., glutathione) may relate to oxidative stress. Furthermore, several secondary metabolic pathways obviously change under K deficiency, including the phenylpropanoid pathway, where accumulated phenylalanine can be converted into some secondary metabolites and salicylic acid (Figure 3), which are critical for stress tolerance. While accumulation of sugar metabolites is observed under deficiency of N, P and K, the increases of soluble sugars may contribute to maintain osmotic homeostasis during nutrient deficiencies. Therefore, both common and specific metabolites or metabolomic pathways can form part in crops response to nutrient deficiency. Although more studies need to be done, some key clues indicate that regulation of C, N and energy metabolisms is important for the responses of crops to nutrient deficiencies, especially regarding macroelements. Elucidating the biosynthesis and regulation of crop metabolites during nutrient deficiency can largely increase our understanding of how plants acquire and utilize mineral nutrients under the fluctuating levels of nutrients in soils. For details on changes made, please see page 19, lines 435-465 in the revised manuscript. In addition, we have increased the font size in all figures and added Figure 3 to summarize the specific metabolomic pathways regulated by low-K stress. For details, please see the figures in the revised manuscript.

  1. Tables 1, 2 and 3 are very little informative. They just collect a number of references and the analytical technique used for the determination of metabolites. This should be corrected by for instance including some info of the research conducted.

Answer: Thank you for your helpful suggestions. We have added some useful information for the metabolic pathways or metabolites significantly regulated by corresponding nutrient deficiency reported by the references in each table. For details on changes made, please see Tables 1, 2 and 3 in the revised manuscript.

  1. Besides the authors, in several occasions, reflect to the “omics” time as an opportunity to deep on the state-of-the-art, but I miss useful information about for instance genetic tools available for genome mining, metabolome mining and so on that can provide useful data to perform holistic studies.

Answer: We have added additional description according to your suggestions. Namely, as an important bridge of genome and phenome, metabolite-based genome-wide association study (mGWAS) has recently been used in interactive functional genomics and metabolomics to understand the genetic bases of plant metabolism [97,98]. The mGWAS approach is performed to identify key genes involved in specific metabolomic pathways in crops. For example, in wheat, several candidate genes are identified as being involved in the flavonoid decoration pathway through mGWAS [99]. Based on mGWAS approach, a genetic network of chlorogenic acid biosynthesis in Populus tomentosa has been constructed by six causal genes [100]. Similar results are also reported in barley for UV-B protection through the regulation of the phenylpropanoid pathway [101]. However, available information about mGWAS used for dissecting mechanisms underlying crop responses to nutrient deficiency is scarce. It is important to identify the critical genes participating in specific metabolomic pathways through integration of mGWAS and other omics approaches, which could be used to develop high nutrient efficiency crop varieties through genetic improvement in future. For details on changes made, please see page 20, lines 485-498 in the revised manuscript.

Reviewer 2 Report

In the manuscript titled, 'Dissection of Crop Metabolome Responses to Nutrient Deficiencies', the authors review recent advancements in the field of metabolomics of plants under nutrient deficiencies. The authors present a clear, well thought out manuscript that superficially describes some of the metabolites that are differentially found in plants under specific nutrient deficiencies. While I like the writing and style of the manuscript, the authors failed to emphasize and give examples of how this type of data can be used to answer important biological questions. I would ask that the authors consider adding one or two paragraphs describing how or what questions they see this information contributing to in the coming years. 

Author Response

  1. In the manuscript titled, 'Dissection of Crop Metabolome Responses to Nutrient Deficiencies', the authors review recent advancements in the field of metabolomics of plants under nutrient deficiencies. The authors present a clear, well thought out manuscript that superficially describes some of the metabolites that are differentially found in plants under specific nutrient deficiencies. While I like the writing and style of the manuscript, the authors failed to emphasize and give examples of how this type of data can be used to answer important biological questions. I would ask that the authors consider adding one or two paragraphs describing how or what questions they see this information contributing to in the coming years.

Answer: We appreciate that you took time to clearly spell out the issues we need to address in the manuscript. According to your helpful comments and suggestions, we have added additional descriptions to discuss the link between macroelement deficiencies and the impact on changes within the metabolic pathways.. Namely, N deficiency seems to impair the whole plant growth, as reflected by decreased N assimilation, TCA cycle and reduced most amino acids (Figure 1), which is considered as an energy-saving strategy for low-N stress tolerance. On the other hand, N deficiency often causes oxidative stress in plants, and thus several stress tolerance-related metabolites, such as galactinol, raffinose, sugar alcohol and ascorbic acid, are accumulated under N-deficient condition (Figure 1), contributing to ROS scavenging [38,39]. Regarding low P availability, N and C metabolisms are also affected by Pi deprivation, along with the TCA cycle and membrane phospholipid metabolism. This can be considered as a fine tune-up to improve P efficiency in plants. For example, increases of sucrose and amino acids in roots seem to support the enlargement of roots. In addition, the reduction of to organic acid metabolites may be attributed to the production of the root exudates to mobilize soil P. Besides, the reduction of phospholipid metabolites, which are important sources of organic P, may contribute to P-reutilization (Figure 2). Different from N and P, K is not a component of most metabolites, and K deficiency resulted in the changes of metabolisms caused by K deficiency may helpful for tolerance to osmotic and oxidative stresses (Figure 3). N metabolism is also regulated by K deficiency, but the changes of amino acids (e.g., glutathione) may relate to oxidative stress. Furthermore, several secondary metabolic pathways obviously change under K deficiency, including the phenylpropanoid pathway, where accumulated phenylalanine can be converted into some secondary metabolites and salicylic acid (Figure 3), which are critical for stress tolerance. While accumulation of sugar metabolites is observed under deficiency of N, P and K, the increases of soluble sugars may contribute to maintain osmotic homeostasis during nutrient deficiencies. Therefore, both common and specific metabolites or metabolomic pathways can form part in crops response to nutrient deficiency. Although more studies need to be done, some key clues indicate that regulation of C, N and energy metabolisms is important for the responses of crops to nutrient deficiencies, especially regarding macroelements. Elucidating the biosynthesis and regulation of crop metabolites during nutrient deficiency can largely increase our understanding of how plants acquire and utilize mineral nutrients under the fluctuating levels of nutrients in soils. For details on changes made, please see page 19, lines 435-465 in the revised manuscript.

  In addition, we also highlighted the metabolite-based genome-wide association study (mGWAS) that can be used for interactive functional genomics and metabolomics to understand the genetic bases of plant metabolism. Namely, as an important bridge of genome and phenome, metabolite-based genome-wide association study (mGWAS) has recently been used in interactive functional genomics and metabolomics to understand the genetic bases of plant metabolism [97,98]. The mGWAS approach is performed to identify key genes involved in specific metabolomic pathways in crops. For example, in wheat, several candidate genes are identified as being involved in the flavonoid decoration pathway through mGWAS [99]. Based on mGWAS approach, a genetic network of chlorogenic acid biosynthesis in Populus tomentosa has been constructed by six causal genes [100]. Similar results are also reported in barley for UV-B protection through the regulation of the phenylpropanoid pathway [101]. However, available information about mGWAS used for dissecting mechanisms underlying crop responses to nutrient deficiency is scarce. It is important to identify the critical genes participating in specific metabolomic pathways through integration of mGWAS and other omics approaches, which could be used to develop high nutrient efficiency crop varieties through genetic improvement in future. For details on changes made, please see page 20, lines 485-498 in the revised manuscript.

Round 2

Reviewer 1 Report

After the first round of revision, the authors managed to review the article accurately. The quality of the figures have improved including useful info in the legend to understand the image. A new figure 3 has been included summarising the K methabolic pathways related to deficiencies of this element. The Tables have also incorporated useful information for the literature review. The authors have also made an effort to impro the content of the "Conclusions and prespectives" section to maximise the impact of the review and provide the expertise vision of the state-of-the-art, useful tools for genome mining, and future steps and opportunities on this research field. In this section there is a correction to be made, the authors refer the Figures, which is not consistent with the structure of a "Conclusions" section, therefore references to the Figures must be removed within this section.

Therefore I consider that the article can be accepted for publication in IJMS.